# PEPNet: A Lightweight Point-based Event Camera 6-DOFs Pose Relocalization Network

## Abstract

Event cameras exhibit remarkable attributes such as high dynamic range, asynchronicity, and low latency, making them highly suitable for vision tasks that involve high-speed motion in challenging lighting conditions. These cameras implicitly capture movement and depth information in events, making them appealing sensors for Camera Pose Relocalization (CPR) tasks. Nevertheless, existing CPR networks based on events neglect the pivotal fine-grained temporal information in events, resulting in unsatisfactory performance. Moreover, the energy-efficient features are further compromised by the use of excessively complex models, hindering efficient deployment on edge devices. In this paper, we introduce PEPNet, a lightweight point-based network designed to regress six degrees of freedom (6-DOFs) event camera poses. We rethink the relationship between the event camera and CPR tasks, leveraging the raw point cloud directly as network input to harness the high-temporal resolution and inherent sparsity of events. PEPNet is adept at abstracting the spatial and implicit temporal features through hierarchical structure and explicit temporal features by Attentive Bi-directional Long Short-Term Memory (A-Bi-LSTM). By employing a carefully crafted lightweight design, PEPNet delivers state-of-the-art (SOTA) performance on public datasets with meager computational resources. Specifically, PEPNet attains a significant 38% performance improvement on the random split DAVIS 240C CPR Dataset, utilizing merely 6% of the parameters compared to traditional frame-based approaches. Moreover, the lightweight design version $PEPNet_{tiny}$ accomplishes results comparable to the SOTA while employing a mere 0.5% of the parameters.

## 1 Introduction

Event cameras are a type of bio-inspired vision sensor that responds to local changes in illumination that exceed a predefined threshold (Lichtsteiner et al., 2008). Differing from conventional frame-based cameras, event cameras independently and asynchronously emit pixel-level events. Notably, event cameras boast an exceptional triad: high dynamic range, low latency, and ultra-high temporal resolution. This unique combination empowers superior performance under challenging light conditions, adeptly capturing the swift scene and rapid motion changes in near-microsecond precision (Posch et al., 2010). Additionally, event cameras boast remarkably low power consumption. Due to their inherent hardware attributes, event cameras have garnered significant attention in the fields of computer vision and robotics in recent years, positioning them as a popular choice for many power-constrained devices like wearable devices, mobile drones, and robots (Delbruck & Lang, 2013; Gallego et al., 2020; Mitrokhin et al., 2019). Camera Pose Relocalization (CPR) is such an example. CPR facilitates the accurate estimation of a camera's pose within the world coordinate system (Sünderhauf et al., 2015). It is extensively employed in numerous applications, including Virtual Reality (VR), Augmented Reality (AR), and robotics (Shavit & Ferens, 2019).

CPR tasks using event cameras significantly diverge from their conventional CPR counterpart that employs frame-based cameras, primarily due to the inherent dissimilarity in data output mechanisms between these two camera types. Furthermore, events inherently encompass information regarding object motion and depth changes across precise temporal and spatial dimensions attributes of paramount significance within the domain of CPR tasks (Rebecq et al., 2018; Gallego

et al., 2017). Regrettably, existing event-based CPR networks often derive from the conventional camera network paradigms and inadequately address the unique attributes of event data. More specifically, events are transformed into various representations such as event images (Nguyen et al., 2019), time surfaces (Lin et al., 2022), and other representations(Lin et al., 2022), leading to the loss of their fine-grained temporal information. Furthermore, most event-based methods tend to overlook the computational load of the network, only prioritizing elevated accuracy, which contradicts the fundamental design principles of event cameras (Gallego et al., 2020).

Point Cloud is a collection of 3D points $(x, y, z)$ that represents the shape and surface of an object or environment and is often used in lidar and depth cameras (Guo et al., 2020). Event Cloud is a collection of events $(x, y, t, p)$ generated by event cameras, $t$ represents timestamps and $p$ is the polarity. By treating each event's temporal information as the third dimension, event inputs $(x, y, t)$ can be transformed into points and aggregated into a pseudo-Point Cloud (Wang et al., 2019; Qi et al., 2017a;b). However, a direct transplantation of the Point Cloud network has not yet exhibited an amazing performance advantage in processing event data. Given that the $t$ dimension of Event Cloud is not strictly equivalent to the spatial dimensions $(x, y, z)$, customizing the Point Cloud network becomes imperative to adequately capture the temporal information of events.

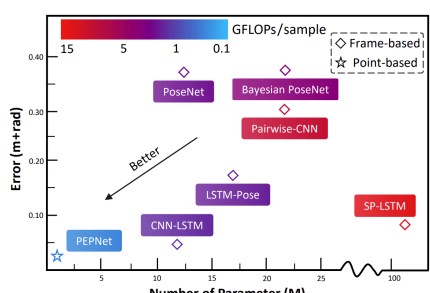

Figure 1: The average results using the random split method benchmarked on the CPR dataset (Mueggler et al., 2017). The vertical axis represents the combined rotational and translational errors (m+rad). PEPNet is the first point-based CPR network for event cameras.

In this study, we introduce PEPNet, an innovative end-to-end CPR network designed to harness the attributes of event cameras. A comparison of our method to other event-based methods is illustrated Figure 2 in in red and blue, respectively. Our main contributions are as follows: First, PEPNet directly processes the raw data obtained from the event cameras, meticulously preserving the fine-grained temporal information and the order inherent in the data. Second, PEPNet proficiently captures spatial features and implicit temporal patterns through its hierarchical structure with temporal aggregation. Additionally, it effectively incorporates explicit temporal features using A-Bi-LSTM. This architecture is tailored to accommodate the high temporal resolution and sparse characteristics inherent in event cameras. Third, PEPNet not only attains SOTA results on a public dataset (Mueggler et al., 2017) but also can be executed in real-time with a lightweight design as shown in Figure 1. Diverging from other point-based approaches in event data processing (Wang et al., 2019; Ren et al., 2023), PEPNet stands out by meticulously considering the distinction between Event Cloud and Point Cloud in its design. This thoughtful approach enables the precise extraction of spatio-temporal features and facilitates solutions for a spectrum of event-based tasks.

## 2 RELATED WORK

### 2.1 FRAME-BASED CPR LEARNING METHODS

Deep learning, crucial for vision tasks like classification and object detection (LeCun et al., 2015), has seen advancements such as PoseNet's innovative transfer learning (Kendall et al., 2015). Utilizing VGG, ResNet (Simonyan & Zisserman, 2014; He et al., 2016), LSTM, and customized loss functions (Walch et al., 2017; Wu et al., 2017; Naseer & Burgard, 2017), researchers enhanced this approach. Auxiliary Learning methods further improved performance (Valada et al., 2018; Radwan et al., 2018; Lin et al., 2019), although overfitting remains a challenge. Hybrid pose-based methods, combining learning with traditional pipelines (Laskar et al., 2017; Balntas et al., 2018), offer promise. DSAC series, for instance, achieve high pose estimation accuracy (Brachmann & Rother, 2021; Brachmann et al., 2017), but come with increased computational costs and latency, especially for edge devices.

### 2.2 EVENT-BASED CPR LEARNING METHODS

Event-based CPR methods often derive from the frame-based CPR network. SP-LSTM (Nguyen et al., 2019) employed the stacked spatial LSTM networks to process event images, facilitating a real-time pose estimator. To address the inherent noise in event images, (Jin et al., 2021) proposed a network structure combining denoise networks, convolutional neural networks, and LSTM, achieving good performance under complex working conditions. In contrast to the aforementioned methods, a novel representation named Reversed Window Entropy Image (RWEI) (Lin et al., 2022) is introduced, which is based on the widely used event surface (Mitrokhin et al., 2020) and serves as

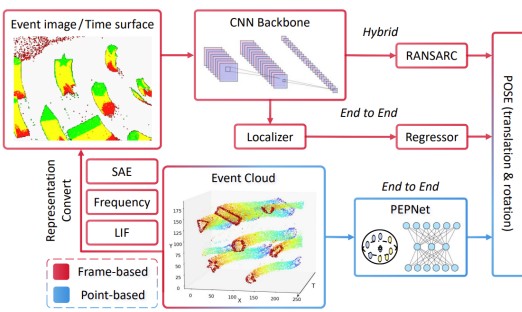

Figure 2: Two different event-based processing methods, frame-based and point-based.

the input to an attention-based DSAC* pipeline (Brachmann & Rother, 2021) to achieve SOTA results. However, the computationally demanding architecture involving representation transformation and hybrid pipeline poses challenges for real-time execution. Additionally, all existing methods ignore the fine-grained temporal feature of the event cameras, and accumulate events into frames for processing, resulting in unsatisfactory performance.

### 2.3 POINT CLOUD NETWORK

Point-based methodologies have transformed the direct processing of Point Cloud, with PointNet (Qi et al., 2017a) as a standout example. Taking a step beyond, PointNet++ (Qi et al., 2017b) introduced a Set Abstraction module. While it initially employed a straightforward MLP in the feature extractor, recent advancements have seen the development of more sophisticated feature extractors to enhance Point Cloud processing (Wu et al., 2019; Zhao et al., 2021; Ma et al., 2021; Dosovitskiy et al., 2020). When extending these techniques to Event Cloud, Wang et al. (Wang et al., 2019) were the first to address the temporal information processing challenge while maintaining representation in both the x and y axes, enabling gesture recognition using PointNet++. Further enhancements came with PAT (Yang et al., 2019), which incorporated self-attention and Gumbel subset sampling, leading to improved performance in recognition tasks. However, existing point-based models still fall short in performance compared to frame-based methods. This phenomenon can be attributed to the distinctively different characteristics of Point Cloud and Event Cloud. Event Cloud contradicts the permutation and transformation invariance present in Point Cloud due to its temporal nature. Additionally, the Point Cloud network is not equipped to extract explicit temporal features.

## 3 PEPNET

PEPNet pipeline consists of four essential modules: (1) a preprocessing module for the original Event Cloud, (2) a hierarchical point cloud feature extraction structure, (3) an Attentive Bidirectional LSTM, and (4) a 6-DOFs pose regressor, as illustrated in Figure 3. In the following sections, we will provide detailed descriptions and formulations for each module.

### 3.1 EVENT CLOUD

To preserve the fine-grained temporal information and original data distribution attributes from the Event Cloud, the 2D-spatial and 1D-temporal event information is constructed into a three-dimensional representation to be processed in Point Cloud. Event Cloud consists of time-series data capturing spatial intensity changes of images in chronological order, and an individual event is denoted as $e_k = (x_k, y_k, t_k, p_k)$, where $k$ is the index representing the $k_{th}$ element in the sequence. Consequently, the set of events within a single sequence ($\mathcal{E}$) in the dataset can be expressed as:

$$\mathcal{E} = \{e_k = (x_k, y_k, t_k, p_k) \mid k = 1, \ldots, n\} \tag{1}$$

For a given pose in the dataset, the ground truth resolution is limited to 5 $ms$, while the event resolution is 1 $\mu s$. Therefore, it is necessary to acquire the events that transpire within the time

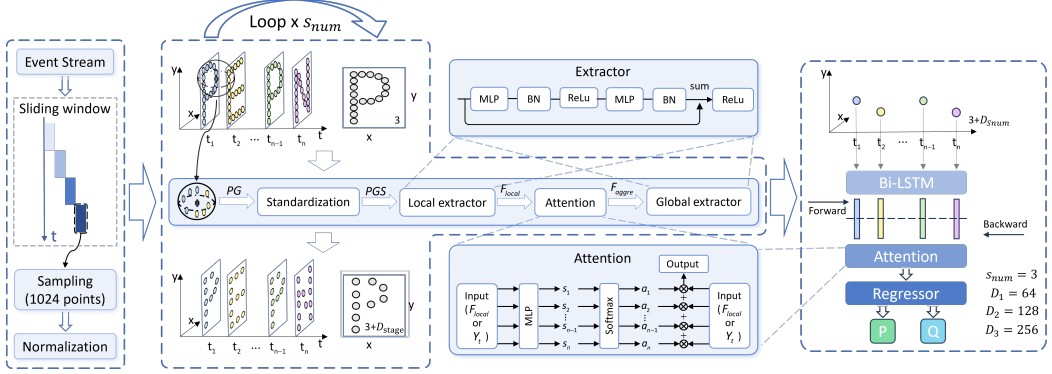

Figure 3: PEPNet overall architecture. The input Event Cloud undergoes direct handling through a sliding window, sampling, and normalization, eliminating the need for any format conversion. Sequentially, the input passes through $S_{num}$ hierarchy structures for spatial feature abstraction and extraction. It further traverses a bidirectional LSTM for temporal feature extraction, culminating in a regressor responsible for 6-DOFs camera pose relocalization.

period we call it sliding window corresponding to the poses, which will serve as the input for the model, as depicted by the following equation:

$$\mathcal{P}_i = \{e_{j \to l} \mid t_l - t_j = R\} \quad i = 1, \dots, M \tag{2}$$

The symbol $R$ represents the time interval of the sliding window, where $j$ and $l$ denote the start and end event index of the sequence, respectively. The variable $M$ represents the number of sliding windows into which the sequence of events $\mathcal{E}$ is divided. Before being fed into the neural network, $P_i$ also needs to undergo sampling and normalization. Sampling is to unify the number of points $N$ as network inputs. We set $N = 1024$ in PEPNet. Additionally, as the spatial coordinates are normalized by the camera's resolution $w$ and $h$. The normalization process is described by the following equation:

$$PN_i = (\frac{X_i}{w}, \frac{Y_i}{h}, \frac{T_i - t_j}{t_l - t_j}) \tag{3}$$

$$X_i, Y_i, T_i = \{x_j, \dots, x_l\}, \{y_j, \dots, y_l\}, \{t_j, \dots, t_l\} \tag{4}$$

The $X, Y$ is divided by the resolution of the event camera. To normalize $T$, we subtract the smallest timestamp $t_j$ of the window and divide it by the time difference $t_l - t_j$, where $t_l$ represents the largest timestamp within the window. After pre-processing, Event Cloud is converted into the pseudo-Point Cloud, which comprises explicit spatial information $(x, y)$ and implicit temporal information $t$.

## 3.2 HIERARCHY STRUCTURE

The hierarchy structure is the backbone for processing the pseudo-3D point cloud and is composed of four primary modules: grouping and sampling, standardization, feature extractor, and aggregation, as described in the following subsection. To efficiently extract deeper explicit spatial and implicit temporal features, the hierarchical structure is tailored and differs from conventional hierarchical structure in a few ways: First, we no longer force permutation invariance as usually done in mainstream point-based methods (Qi et al., 2017a; Ma et al., 2021), as the motion information is inherently related to the sequential order of events. Instead, we **keep the sequence of all events strictly in the same order** as they are generated to preserve the temporal information to be used in the next stage. Second, we replace MaxPooling in aggregation and deploy temporal aggregation which leverages the attention mechanism with softmax, which improves the effective assimilation of temporal information into the resultant feature vectors.

### 3.2.1 GROUPING AND SAMPLING

Aligned with the frame-based design concept, our focus is to capture both local and global information. Local information is acquired by leveraging Farthest Point Sampling (FPS) and K-Nearest Neighbors (KNN), while global information is obtained through a dedicated aggregation module.

$$PS_i = FPS(PN_i) \quad PG_i = KNN(PN_i, PS_i) \tag{5}$$

The input dimension $PN_i$ is $[N, 3 + D]$, and the centroid dimension $PS_i$ is $[N', 3 + D]$ and the group dimension $PG_i$ is $[N', K, 3 + 2 * D]$. $K$ represents the nearest $K$ points of the center point

(centroid), D is the feature dimension of the points of the current stage, and 3 is the most original $(X, Y, T)$ coordinate value. Importantly, it should be noted that the ordering of all points in the grouping and sampling process strictly adheres to the timestamp (T).

### 3.2.2 STANDARDIZATION

Next, each group undergoes a standardization process to ensure consistent variability between points within the group, as illustrated in this formula:

$$PGS = \frac{PG - PS}{Std(PG)} \quad Std(PG_i) = \sqrt{\frac{\sum_{j=0}^{3n-1}(g_j - \bar{g})^2}{3n - 1}} \tag{6}$$

$$g = [x_0, y_0, t_0, \ldots, x_n, y_n, t_n] \tag{7}$$

Where $PG_i$ and $PS_i$ are the subsets of $PG$ and $PS$, $Std$ is the standard deviation, the dimension of $Std(PG)$ is $[M]$ which is consistent with the number of sliding windows, and $g$ is the set of coordinates of all points in the $PG_i$.

### 3.2.3 FEATURE EXTRACTOR

Following the standardization of $PG$ by dividing the variance by the subtracted mean, the feature extraction is performed using a Multi-Layer Perceptron (MLP) with a residual connection. This process encompasses two steps: local feature extraction and global feature extraction. The feature extractor with a bottleneck can be mathematically represented as:

$$I(x) = f(BN(MLP_1(x))) \tag{8}$$

$$O(x) = BN(MLP_2(x)) \tag{9}$$

$$Ext(x) = f(x + O(I(x))) \tag{10}$$

$BN$ represents batch normalization layer, while $f$ signifies the nonlinear activation function. Both local feature extraction and global feature extraction maintain identical input and output dimensions. The dimension increase occurs solely when combining the feature dimension D of the current point with the feature dimension $D$ of the centroid during grouping, resulting in a final dimension of $2 * D$. The feature extractor takes an input dimension of $[B, N, K, D]$, and following local feature extraction, the dimension remains $[B, N, K, D]$, $B$ represents batch size. We adopt the attention mechanism for aggregation, yielding an aggregated feature dimension of $[B, N, D]$. Subsequently, the aggregated feature map of $[B, N, D]$ is then processed through the global feature extractor, completing the feature extraction for the current stage.

### 3.2.4 TEMPORAL AGGREGATION

Conventional Point Cloud methods favor MaxPooling operations for feature aggregation because it is efficient in extracting the feature from one point among a group of points and discarding the rest. However, MaxPooling involves extracting only the maximum value along each dimension of the temporal axis. It is robust to noise perturbation but also ignores the temporal nuances embedded within the features. Conversely, the integration of attention mechanisms enhances the preservation of those nuanced but useful temporal attributes by aggregating features along the temporal axis through the attention value. To provide a more comprehensive exposition, we employ a direct attention mechanism within the $K$ temporal dimensions to effectively aggregate features as shown in Figure 3. This mechanism enables the explicit integration of temporal attributes, capitalizing on the inherent strict ordering of the $K$ points. The ensuing formula succinctly elucidates the essence of this attention mechanism:

$$F_{local} = Ext(x) = (S_{t1}, S_{t2}, \ldots, S_{tk}) \tag{11}$$

$$A = SoftMax(MLP(F_{local})) = (a_{t1}, a_{t2}, \ldots, a_{tk}) \tag{12}$$

$$F_{aggre} = A \cdot F_{local} = S_{t1} \cdot a_{t1} + S_{t2} \cdot a_{t2} + \cdots + S_{tk} \cdot a_{tk} \tag{13}$$

Upon the application of the local feature extractor, the ensuing features are denoted as $F_{local}$, and $S_{tk}$ mean the extracted feature of $k_{th}$ point in a group. The attention mechanism comprises an MLP layer with an input layer dimension of $D$ and an output $a_{tk}$ dimension of 1, along with softmax layers. Subsequently, the attention mechanism computes attention values, represented as $A$. These attention values are then multiplied with the original features through batch matrix multiplication, resulting in the aggregated feature $F_{aggre}$.

## 3.3 A-Bi-LSTM

The temporal features extracted through the hierarchical structure are independent and parallel, lacking recurrent mechanisms within the network. This distinctive attribute, referred to as 'implicit', contrasts with the conventional treatment of temporal information as an indexed process. Consequently, implicit temporal features **inadequately capture the interrelations among events along the timeline**, whereas explicit temporal features assume a pivotal role in facilitating the CPR task. To explicitly capture temporal patterns, we introduce the LSTM network, which has been proven effective in learning temporal dependencies. For optimal network performance, controlled feature dimensionality, and comprehensive capture of bidirectional relationships in pose context, we adopt a bi-directional LSTM network with a lightweight design. The integration of bidirectional connections into the recurrent neural network (RNN) is succinctly presented through the following equation:

$$\mathbf{h}_t = f(\mathbf{W}_h \cdot \mathbf{x}_t + \mathbf{U}_h \cdot \mathbf{h}_{t-1} + \mathbf{b}_h) \tag{14}$$

$$\mathbf{h}'_t = f(\mathbf{W}'_h \cdot \mathbf{x}_t + \mathbf{U}'_h \cdot \mathbf{h}'_{t+1} + \mathbf{b}'_h) \tag{15}$$

$$\mathbf{y}_t = \mathbf{V} \cdot \mathbf{h}_t + \mathbf{b}_y \tag{16}$$

$$\mathbf{y}'_t = \mathbf{V}' \cdot \mathbf{h}'_t + \mathbf{b}'_y \tag{17}$$

$\mathbf{x}_t$ represents the feature vector at the $t$-th time step of the input sequence, while $\mathbf{h}_{t-1}$ and $\mathbf{h}'_{t+1}$ correspond to the hidden states of the forward and backward RNN units, respectively, from the previous time step. The matrices $\mathbf{W}_h$, $\mathbf{U}_h$, and $\mathbf{b}_h$ denote the weight matrix and bias vector of the forward RNN unit, while $\mathbf{V}$ and $\mathbf{b}_y$ represent the weight matrix and bias vector of its output layer. Similarly, $\mathbf{W}'_h$, $\mathbf{U}'_h$, and $\mathbf{b}'_h$ are associated with the weight matrix and bias vector of the backward RNN unit, and $\mathbf{V}'$ and $\mathbf{b}'_y$ pertain to the weight matrix and bias vector of its output layer. The activation function, denoted as $f(\cdot)$, can be chosen as sigmoid or tanh or other functions. The final output $Y_a$ is aggregated at each moment using the attention mechanism, and $\oplus$ means concat operation.

$$Y_t = y_t \oplus y'_t \tag{18}$$

$$A = SoftMax(MLP(Y_t)) \tag{19}$$

$$Y_a = A \cdot Y_t \tag{20}$$

## 3.4 LOSS FUNCTION

A fully connected layer with a hidden layer is employed to address the final 6-DOFs pose regression task. The displacement vector of the regression is denoted as $\hat{p}$ representing the magnitude and direction of movement, while the rotational Euler angles are denoted as $\hat{q}$ indicating the rotational orientation in three-dimensional space.

---

**Algorithm 1** PEPNet pipeline

**Input**: Raw Event Cloud $\mathcal{E}$
**Parameters**: $N_p = 1024, R = 5e + 3, S_{num} = 3$
**Output**: 6-DOFs pose $(\hat{p}, \hat{q})$

1: **Preprocessing**
2: **for** $j$ **in** len($\mathcal{E}$) **do**
3:    $P_i.append(e_{j \rightarrow l})$ ; $j = l$; where $t_l - t_j = R$
4:    **if** $(len(P_i) > N_p)$: $i = i + 1$;
5: **end for**
6: $PN = Normalize(Sampling(P))$
7:
8: **Hierarchy structure**
9: **for** stage in $range(S_{num})$ **do**
10:    **Grouping and Sampling**$(PN)$
11:    Get $PGS \in [B, N_{stage}, K, 2 * D_{stage-1}]$
12:    **Local Extractor**$(PGS)$
13:    Get $F_{local} \in [B, N_{stage}, K, D_{stage}]$
14:    **Attentive Aggregate**$(F_{local})$
15:    Get $F_{aggre} \in [B, N_{stage}, D_{stage}]$
16:    **Global Extractor**$(F_{aggre})$
17:    Get $PN = F_{global} \in [B, N_{stage}, D_{stage}]$
18: **end for**
19:
20: **A-Bi-LSTM**
21: Forward Get $y_t \in [B, N_3, D_{S_{num}}/2]$
22: Reverse Get $y'_t \in [B, N_3, D_{S_{num}}/2]$
23: Attention Get $Y_a \in [B, D_{S_{num}}]$
24:
25: **Regressor**
26: Get 6-DOFs pose $(\hat{p}, \hat{q})$

---

$$Loss = \alpha||\hat{p} - p||_2 + \beta||\hat{q} - q||_2 + \lambda\sum_{i=0}^{n}w_i^2 \tag{21}$$

$p$ and $q$ represent the ground truth obtained from the dataset, while $\alpha$, $\beta$, and $\lambda$ serve as weight proportion coefficients. In order to tackle the prominent concern of overfitting, especially in the end-to-end setting, we propose the incorporation of L2 regularization into the loss function. This regularization, implemented as the second paradigm for the network weights $w$, effectively mitigates the impact of overfitting.



Figure 4: Event-based CPR Dataset visualization.

## 3.5 OVERALL ARCHITECTURE

Next, we will present the PEPNet pipeline in pseudo-code, utilizing the previously defined variables and formulas as described in Algorithm 1.

## 4 EXPERIMENT

In this section, we present an extensive and in-depth analysis of PEPNet's performance on a public dataset, encompassing evaluations based on rotational and translational mean squared error (MSE), model parameters, floating-point operations (FLOPs), and inference time. Through a series of systematic ablation experiments, we experimentally validate the efficacy of each module. PEPNet's training and testing are performed on a server furnished with an AMD Ryzen 7950X CPU, an RTX GeForce 4090 GPU, and 32GB of memory.

### 4.1 DATASET

We employ the widely evaluated event-based CPR dataset (Mueggler et al., 2017) collected using the DAVIS 240C. This dataset encompasses a diverse set of multimodal information, comprising events, images, IMU measurements, camera calibration, and ground truth information acquired from a motion capture system operating at an impressive frequency of 200 Hz, thereby ensuring sub-millimeter precision. We visualized various types of sequences as shown in Figure 4.

Two distinct methods to partition the dataset (Nguyen et al., 2019) have been benchmarked: random split and novel split. In the random split approach, the dataset is randomly selected 70% of all sequences for training and allocated the remaining sequences for testing. On the other hand, in the novel split, we divide the data chronologically, using the initial 70% of sequences for training and the subsequent 30% for testing.

### 4.2 BASELINE

We perform a thorough evaluation of our proposed method by comparing it with SOTA event-based approaches, namely CNN-LSTM (Tabia et al., 2022) and AECRN (Lin et al., 2022). Moreover, we present results derived from other well-established computer vision methods, including PoseNet(Kendall et al., 2015), Bayesian PoseNet (Kendall & Cipolla, 2016), Pairwise-CNN (Laskar et al., 2017), LSTM-Pose (Walch et al., 2017), and SP-LSTM(Nguyen et al., 2019).

### 4.3 RANDOM SPLIT RESULTS

Based on the findings presented in Table 1, it is apparent that PEPNet surpasses other models concerning both rotation and translation errors across all sequences. Notably, PEPNet achieves these impressive results despite utilizing significantly fewer model parameters and FLOPs compared to the frame-based approach. Moreover, PEPNet not only exhibits a remarkable 38% improvement in the average error compared to the SOTA CNN-LSTM method but also attains superior results across nearly all sequences.In addressing the more intricate and challenging hdr_poster sequences, while the frame-based approach relies on a denoising network to yield improved results (Jin et al., 2021), PEPNet excels by achieving remarkable performance without any additional processing. This observation strongly implies that PEPNet's point cloud approach exhibits greater robustness compared to the frame-based method, highlighting its inherent superiority in handling complex scenarios.

Furthermore, we introduce an alternative variant, PEPNet$_{tiny}$, which integrates a lighter model architecture while preserving relatively strong performance. As depicted in Figure 3, PEPNet consists

| Network | PoseNet | Bayesian PoseNet | Pairwise-CNN | LSTM-Pose | SP-LSTM | CNN-LSTM | PEPNet | PEPNet$_{tiny}$ |
|---|---|---|---|---|---|---|---|---|
| Parameter | 12.43M | 22.35M | 22.34M | 16.05M | 135.25M | 12.63M | 0.774M | **0.064M** |
| FLOPs | 1.584G | 3.679G | 7.359G | 1.822G | 15.623G | 1.960G | 0.459G | **0.033G** |
| shapes_rotation | 0.109m,7.388° | 0.142m,9.557° | 0.095m,6.332° | 0.032m,4.439° | 0.025m,2.256° | 0.012m,1.652° | **0.005m,1.372°** | 0.006m,1.592° |
| box_translation | 0.193m,6.977° | 0.190m,6.636° | 0.178m,6.153° | 0.083m,6.215° | 0.036m,2.195° | **0.013m**,0.873° | 0.017m,**0.845°** | 0.031m,1.516° |
| shapes_translation | 0.238m,6.001° | 0.264m,6.235° | 0.201m,5.146° | 0.056m,5.018° | 0.035m,2.117° | 0.020m,1.471° | **0.011m,0.582°** | 0.013m, 0.769° |
| dynamic_6dof | 0.297m,9.332° | 0.296m,8.963° | 0.245m,5.962° | 0.097m,6.732° | 0.031m,2.047° | 0.016m,1.662° | **0.015m,1.045°** | 0.018m,1.144° |
| hdr_poster | 0.282m,8.513° | 0.290m,8.710° | 0.232m,7.234° | 0.108m,6.186° | 0.051m,3.354° | 0.033m,2.421° | **0.016m,0.991°** | 0.028m,1.863° |
| poster_translation | 0.266m,6.516° | 0.264m,5.459° | 0.211m,6.439° | 0.079m,5.734° | 0.036m,2.074° | 0.020m,1.468° | **0.012m,0.588°** | 0.019m,0.953° |
| Average | 0.231m,7.455° | 0.241m,7.593° | 0.194m,6.211° | 0.076m,5.721° | 0.036m,2.341° | 0.019m,1.591° | **0.013m,0.904°** | 0.019m,1.306° |

Table 1: Random split results. The table presents the median error for each sequence, as well as the average error across the six sequences. It also presents the number of parameters and FLOPs for each model. Bold indicates the most advanced result, while underline signifies the second-best result.

| Network | PoseNet | Bayesian PoseNet | Pairwise-CNN | LSTM-Pose | SP-LSTM | DSAC* | AECRN | **PEPNet** |
|---|---|---|---|---|---|---|---|---|
| shapes_rotation | 0.201m,12.499° | 0.164m,12.188° | 0.187m,10.426° | 0.061m,7.625° | 0.045m,5.017° | 0.029m,2.3° | 0.025m,2.0° | **0.016m,1.745°** |
| shapes_translation | 0.198m,6.696° | 0.213m,7.441° | 0.225m,11.627° | 0.108m,8.468° | 0.072m,4.496° | 0.038m,2.2° | 0.029m,1.7° | **0.026m,1.659°** |
| shapes_6dof | 0.320m,13.733° | 0.326m,13.296° | 0.314m,13.245° | 0.096m,8.973° | 0.078m,5.524° | 0.054m,3.1° | 0.052m,3.0° | **0.045m,2.984°** |
| Average | 0.240m,11.067° | 0.234m,10.975° | 0.242m,11.766° | 0.088m,8.355° | 0.065m,5.012° | 0.040m,2.53° | 0.035m,2.23° | **0.029m,2.13°** |
| Inference time | 5ms | 6ms | 12ms | 9.49ms | 4.79ms | 30ms | 30ms | **6.7ms** |

Table 2: Novel split results. Referred to as Table I, showcases identical information. To assess the model's runtime, we conduct tests on a server platform, specifically focusing on the average time required for inference on a single sample.

of three stages, and the model's size is contingent upon the dimensionality of MLPs at each stage. The dimensions for the standard structure are [64, 128, 256], whereas those for the tiny structure are [16, 32, 64]. As indicated in Table 1, even with a mere 0.5% of the CNN-LSTM's parameter, PEPNet$_{tiny}$ achieves comparable and even slightly superior results. This remarkable outcome emphasizes the superiority of leveraging event cloud data processing directly.

Although PEPNet$_{tiny}$ demonstrates the potential to outperform previous SOTA results in terms of the final average performance, it reveals evident weaknesses and underfitting when handling more complex sequences, such as hdr_poster and box_translation. The limitations in the abstraction ability of PEPNet$_{tiny}$ become apparent. It is important to acknowledge that PEPNet's results might improve with a larger dataset, indicating the significant impact of data size on the model's performance.

### 4.4 ERROR DISTRIBUTION

Figure 5 illustrates the error distribution of

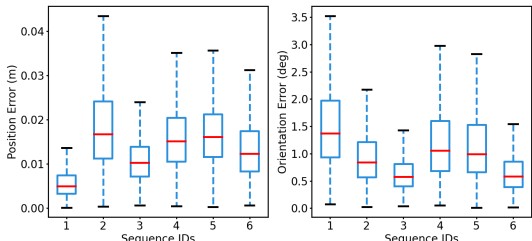

Figure 5: Error distribution of event-based CPR results achieved by PEPNet using a random split. (a) Translation errors. (b) Rotation errors.

PEPNet across six distinct sequences using the random split method, specifically: shape rotation, box translation, shape translation, dynamic 6-dof, hdr poster, and poster translation. To enhance clarity, the top and bottom boundaries of the box represent the first and third quartiles, respectively, indicating the inter-quartile range (IQR). The median is denoted by the band within the box. It is observed that the IQR of the translation error approximately locates between 0.004m and 0.024m, while the orientation error ranges from 0.4° to 1.9°.

Among the six sequences, shape rotation and box translation display the poorest results in rotation and translation, respectively, primarily due to the inherent complexity of the dataset. As the scene becomes more intricate and the resolution increases, such as in the hdr poster, the model is challenged to exhibit its robustness. Notably, PEPNet demonstrates enhancements of approximately 50% compared to the SOTA model in this scenario.

### 4.5 NOVEL SPLIT RESULTS

To assess the model's robustness, we adopt the novel split as an evaluation criterion, as shown in Table 2. During the training process, we observe a more pronounced overfitting phenomenon in PEPNet compared to the random split. We attribute this observation to the disparities in data distributions between the trainset and the testset, as well as the limited data size. Contrary to the

| Condition | HS | LSTM | Bi-LSTM | Aggregation | Translation | Rotation | T+R |
|:---:|:---:|:---:|:---:|:---:|:---:|:---:|:---:|
| 1 | ✓ | | | Max | 0.015m | 0.884° | 3.04 |
| 2 | ✓ | | | Temporal | 0.014m | 0.786° | 2.77 |
| 3 | ✓ | ✓ | | Max | 0.014m | 0.833° | 2.85 |
| 4 | ✓ | | ✓ | Max | 0.014m | 0.813° | 2.82 |
| 5 | ✓ | | ✓ | Temporal | **0.011m** | **0.582°** | **2.12** |

Table 3: Abalation Study for three key modules. T+R = Translation + Rotation·$\pi/180$ (m+rad)

methods we compared, PEPNet does not necessitate pre-trained weights. For instance, SP-LSTM relies on pre-trained VGG19 weights from Imagenet, while AECRN requires synthetic heuristic depth and an extensive pretraining process.

To address overfitting, PEPNet employs conventional methods that yield consistent and comparable results with the SOTA on three shape sequences that are displayed in the network column of Table 2. It is essential to note that AECRN adopts a hybrid approach, combining neural network regression for scene coordinates with derivable RANSAC for pose estimation. Moreover, this method incurs significant time consumption, with even the SOTA DSAC* algorithm taking nearly 30ms, excluding additional time for format conversion. This time constraint presents compatibility challenges with the low-latency nature of event cameras. In contrast, PEPNet can execute on a server in just 6.7ms, with the main time-consuming module being grouping and sampling. Furthermore, with potential field programmable gate array (FPGA) or application-specific integrated chip (ASIC) support for these operations, PEPNet's performance can be further accelerated.

### 4.6 ATTENTION VISUALIZATION

As shown in Figure 6, We observe that the values exhibit larger at both the start and end. Our conjecture posits that during the process of camera pose relocalization, the model may intensify its emphasis on the distinctions in features between the initial and terminal points, and regress the 6DOFs pose through the differences, similar to geometric methods Mueggler et al. (2018); Gallego et al. (2015).

### 4.7 ABLATION STUDY

In order to validate the efficacy of key modules, we conducted ablation experiments focusing on three primary components: hierarchy structure, Bi-LSTM, and attention. These experiments are designed to evaluate rotation and translation errors on the shape translation sequence with random split. The combined error (T+R) is measured after processing.

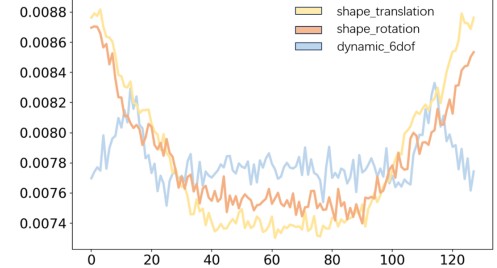

Figure 6: Visualization of the attention values in the time domain. 128 points in chronological order on the horizontal axis and the attention values of the corresponding point on the vertical axis.

Our experimental setup comprises four distinct conditions, as illustrated in Table 3. Condition 1 represents the sole utilization of the hierarchy structure (HS), while Condition 2 combines the ordinary LSTM. Condition 3 incorporates the bidirectional LSTM, and Condition 4 integrates the attention mechanism for feature aggregation.

The ablation experiments reveal significant insights. Experiments 1 and 2 demonstrate that augmenting LSTM enhances the extraction of explicit temporal features. Moreover, experiments 2 and 3 reveal the effectiveness of the bidirectional LSTM in extracting motion information. Additionally, experiments 3 and 4 confirm the notable impact of attention in feature aggregation, resulting in a substantial reduction in error rates.

### 5 CONCLUSION

In this paper, we introduce an end-to-end CPR network that operates directly on raw event clouds without frame-based preprocessing. PEPNet boasts an impressively lightweight framework that adeptly extracts spatial and temporal features, leading to SOTA outcomes on publicly accessible datasets. Diverging from traditional frame-based approaches, our method prioritizes preserving the inherent distribution of the event camera output, capitalizing on its sparse nature to achieve extraordinary capabilities for ultra-low-power CPR applications.

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
