# OpenReview forum: "PEPNet: A Lightweight Point-based Event Camera 6-DOFs Pose Relocalization Network"
_ICLR.cc/2024/Conference — ICLR 2024 Conference Withdrawn Submission_

### Official Review · Reviewer_ofst · 2023-10-23

**Soundness:** 3 good
**Presentation:** 3 good
**Contribution:** 3 good
**Rating:** 5
**Confidence:** 4

**Summary:**

This paper introduces a lightweight point-based end-to-end network (PEPNet) for camera relocalization from event cameras.  The network extracts spatial and implicit temporal features through a hierarchical structure and then explicitly attains temporal information via an attentive bi-directional LSTM (A-Bi-LSTM).  Experimental results on the DAVIS 240C CPR dataset show its good performance as well as the effacacy.

**Strengths:**

1) The way to treat the time dimension for event cameras in a point-based network is insightful.

2) The resultant network is lightweight yet performs well on the benchmarking dataset.

**Weaknesses:**

-- The symbol for concatenation operation is ambiguous. The same symbol is also used in Fig. 3 (Extractor), but it seems it does not mean concatenation there.

-- Fig. 4 can be replaced with qualitative comparison to the baselines rather than just some existing data samples from the dataset.

-- Some citations for the baselines are missing in Section 4.2.

**Questions:**

1) Section 3.2.2: What are the dimensions of PG and PS? How is the substraction done? Could the authors elaborate this in detail?

2) Section 3.2.3: Does f correspond with the ReLu operator in Fig. 3? It seems there is no ReLu before the summation operator in Fig. 3, but there is an f in Eq. (9). Why are they different?

3) Are the random data train/test splits the same as the baselines? "We randomly select" sounds like the authors use a different random split although following the same strategy. I wonder whether the comparison on this setting is really fair or not.

4) How is PEPNet_{tiny} obtained exactly?

---

> ### Author Response · Authors · 2023-11-13
> **We appreciate your recognition of the PEPNet’s way to treat the time dimension. We would like to address your questions with the utmost diligence and clarity.**
>
> **Weakness 1. The symbol for the concatenation operation is ambiguous. The same symbol is also used in Fig. 3 (Extractor), but it seems it does not mean concatenation there.**
>
> We apologize for the ambiguity due to the same symbols in Equation 18 and Figure 3. We have modified the main figure in the revised paper to eliminate the ambiguity in the Fig and Equation.
>
> **Weakness 2. Fig. 4 can be replaced with a qualitative comparison to the baselines rather than just some existing data samples from the dataset.**
>
> Our initial goal was to visualize different types of data to help readers better understand the event-based task of CPR. Unfortunately, CPR tasks may not have a good visualization method to show the comparison with the baseline, and the most intuitive enhancement is the reduction of MSE. In more detail, the pose is not visualized in a good way and the error variance (from centimeter to meter) of the predictions of the different models from the ground truth is large.   (Unlike detection and segmentation tasks which can visualize how good the results are).
>
> **Weakness 3. Some citations for the baselines are missing in Section 4.2.**
>
> In the revised version, we have provided the previously missing citations in line 296.
>
> **Q1. Section 3.2.2: What are the dimensions of PG and PS? How is the subtraction done? Could the authors elaborate on this in detail?**
>
> We should have been clearer about this. The dimensions of $PG_i$ and $PS_i$ are defined in Section 3.2.1.  $PG_i$ and $PS_i$ are subsets of PG and PS in the $i^{th}$ sliding window, respectively.  PG represents a set of a group of Point Cloud with dimension [M, N', K, D+3] after grouping operation. PS is the centroid data of each group of dimension [M, N', D+3]， M is the number of sliding windows, N' is the number of groups in the point cloud, K represents the number of points in each group, and D is the dimension of each point.  The operation of PG-PS firstly replicates the PS in the third dimension K times into [M, N', K, D+ 3], and then does the phase reduction operation. To avoid confusion to readers, we have modified the description of Eq. 6 and Eq.7 in line 183.
>
> **Q2. Section 3.2.3: Does f correspond with the ReLu operator in Fig. 3? It seems there is no ReLu before the summation operator in Fig. 3, but there is an f in Eq. (9). Why are they different?**
>
> We sincerely apologize for the error in the formula and have rectified it in the revised version. Additional information describing f has been included in line 191.
>
> **Q3. Are the random data train/test splits the same as the baselines? "We randomly select" sounds like the authors use a different random split although following the same strategy. I wonder whether the comparison of this setting is really fair or not.**
>
> Our dataset processing code is derived from the event-based CPR pioneer [1], and we ensure baseline consistency. We have modified the manuscript to make this clearer in line 287.
>
> [1] Nguyen A, Do T T, Caldwell D G, et al. Real-time 6dof pose relocalization for event cameras with stacked spatial lstm networks[C]//Proceedings of the IEEE/CVF Conference on Computer Vision and Pattern Recognition Workshops. 2019: 0-0.
>
> **Q4. How is PEPNet_{tiny} obtained exactly?**
>
> As illustrated in Figure 3, PEPNet comprises three stages, and the size of the model is determined by the dimensionality of the MLP at each stage. The dimensions for the regular structure are [64,128,256], while the dimensions for the tiny structure are [16,32,64]. A detailed description of these dimensions will be incorporated into the main paper.
>
> Lastly, we express our gratitude again for your insightful comments on the specific details highlighted in our paper, which hold significant importance for us.

---

### Official Review · Reviewer_ndwh · 2023-11-01

**Soundness:** 3 good
**Presentation:** 3 good
**Contribution:** 3 good
**Rating:** 6
**Confidence:** 4

**Summary:**

This paper focuses on the regression task of six degrees of freedom event camera poses. Considering that the existing methods neglect fine-grained temporal information in events, the proposed PEPNet directly deal with the raw point cloud to use the high-temporal resolution and inherent sparsity of events.

**Strengths:**

The proposed PEPNet outperforms other methods in both performance and running speed.

**Weaknesses:**

Weakness:

1.	The event camera data in the method is actually more like video data (not 3-D spatial data as point clouds), probably other video-based feature extraction methods could be utilized for the task.

2.	The proposed hierarchy structure, Bi-LSTM, and attention mechanisms are very common modules in the area of CNNs models for processing video data. This reduces the contributions of the method a little. However, the method does achieve better performance than previous methods.

3.	Is there any ablation study on the loss term weight?

4.	Are there intermediate visualization results of the attention mechanisms to show what has been learned in the module?

5.	Some typos: Second line of P6.

**Questions:**

As above.

---

> ### Author Response · Authors · 2023-11-16
> **We appreciate your acknowledgment of our work and would like to address your inquiry regarding weaknesses.**
>
> **Weakness 1. Other video-based feature extraction methods could be utilized for the task.**
>
> We agree with the reviewer that some video-based methods could be potentially used for the task. However,  as the event camera features a very high effective frame rate, if one would perform the task within the moderate computational effort, employing other video-based methods requires significant downsampling and the temporal accuracy cannot be maintained [1]. Figure 3 depicts the carton form of the point cloud data, maintaining the event’s original time resolution of 1us or equivalently 1,000,000 Hz. This high temporal resolution surpasses the 60-120 Hz frame rate of mainstream video captured by traditional cameras. The computational and processing demands become immeasurable when employing conventional video feature extraction methods due to this stark contrast in resolutions.
>
> [1] Wang Q, Zhang Y, Yuan J, et al. Space-time event clouds for gesture recognition: From RGB cameras to event cameras[C]//2019 IEEE Winter Conference on Applications of Computer Vision (WACV). IEEE, 2019: 1826-1835.
>
> **Weakness 2. Common modules in the area of CNN models for processing video data.**
>
> Indeed, Bi-LSTM and attention mechanisms have been adapted in many CNN models. Our network may appear in the form of common modules, but it is the result of restricting ourselves to minimal hardware resources. Our primary goal is to democratize computer vision technology by making it accessible to a wider range of devices and applications in the community of edge computing. Therefore, we restrict ourselves to modules with only basic and low computational resource requirements.  We strive to improve network performance as much as possible with limited resources and only basic operations.
>
> Recently, there have been many modern innovations in architecture, e.g., GAN, VAE, and vision transformers.  Moreover, those advanced architecture requires high-demand MAC operations and a heavy memory footprint.  Moreover, we outperform all the methods on the event-based CPR tasks,
>
> **Weakness 3. Is there any ablation study on the loss term weight?**
>
> | Scence | $\alpha$=0.5,$\beta$=0.5 | $\alpha$=0.25,$\beta$=0.75 | $\alpha$=0.75,$\beta$=0.25 |
> | -------- | -------- | -------- | -------- |
> | shape\_translation     | **0.0302m,1.684$^\circ$,5.96**   | 0.0359m,1.72$^\circ$,6.59    | 0.0303m,2.056$^\circ$,6.62    |
> | shape\_rotation    | 0.0143m,2.888$^\circ$,6.47     | **0.0159m,2.68$^\circ$,6.27**  | 0.014m,3.36$^\circ$,7.26     |
> | dynamic\_6dof   | **0.0542m,2.799$^\circ$,10.3**  | 0.0611m,2.488$^\circ$,10.5     | 0.0516m,3.251$^\circ$,10.8     |
>
>
> we have incorporated ablation experiments involving correlation coefficients of the loss function.  These experiments utilized a tiny version of PEPNet, trained for 100 epochs.  Across three distinct motion scenarios (translation, rotation, and 6dof) varied coefficient ratios induced deviations in the obtained results.  For example, in shape rotation, increasing the weight on rotation makes the results better.  However, all of our experiments were based on $\alpha$/$\beta$=1, and we did not adjust the ratios for different scenarios to verify the generalization ability of the model.
>
> **Weakness 4. Are there intermediate visualization results of the attention mechanisms to show what has been learned in the module?**
>
> We sincerely appreciate the insightful suggestion you've provided, which has enabled us to have a deeper understanding of the reasons for PEPNet's superior performance.  We visualized the values of temporal attention in Figure 6 in the rebuttal version, with 128 points in chronological order on the horizontal axis and the attention values of the corresponding point on the vertical axis. We discovered that the attention scores at the beginning and the end are higher. We tentatively infer that the model focuses more on the difference in features between the start and the end for Camera Pose Relocalization, which is also seen in the geometry approach [1][2].
>
> [1] Gallego G, Forster C, Mueggler E, et al. Event-based camera pose tracking using a generative event model[J]. arXiv preprint arXiv:1510.01972, 2015.
>
> [2] Mueggler E, Gallego G, Rebecq H, et al. Continuous-time visual-inertial odometry for event cameras[J]. IEEE Transactions on Robotics, 2018, 34(6): 1425-1440.
>
> Finally, we would like to express our gratitude for your time and effort once again.

---

### Official Review · Reviewer_9ZiA · 2023-11-05

**Soundness:** 2 fair
**Presentation:** 3 good
**Contribution:** 2 fair
**Rating:** 3
**Confidence:** 4

**Summary:**

The paper presents a lightweight DNN for pose estimation using event-based camera data. The evaluation is done on a DAVIS-240C camera dataset that contains a variety of camera motions and egomotion ground truth.

**Strengths:**

The paper is clearly written and illustrations support the text well. If the code for this work is released, this would be a notable contribution, as a lightweight method could be readily used in robotic applications.

**Weaknesses:**

1) The literature overview does not include a few works on event-based pose estimation - there are methods based on 3D pointcloud analysis (https://openaccess.thecvf.com/content_CVPR_2020/papers/Mitrokhin_Learning_Visual_Motion_Segmentation_Using_Event_Surfaces_CVPR_2020_paper.pdf), and self-supervised methods (https://arxiv.org/pdf/1903.07520.pdf) - and more, all of which were evaluated on more complex datasets than in this paper. How would these approaches (to DNN, loss functions, event encoding) compare with this work?

2) CPR Dataset evaluated in the paper contains mostly planar scenes or one-dimensional motion, making it easier for the network to overfit. It is also hard to gauge the full 6dof performance of the method, when using simplistic data. I would recommend evaluating on MVSEC (https://daniilidis-group.github.io/mvsec/) and/or EV-IMO (https://better-flow.github.io/evimo/download_evimo_2.html) - both datasets have pose ground truth.

3) Since the problem of motion estimation is geometric, it would benefit the method if the loss function incorporated some geometry constraints. In classic vision, sota egomotion pipelines leverage this successfully, and there were prior works on event cameras doing the same.

4) On motion estimation problem, with a random split, it is highly likely the network overfitted.

**Questions:**

1) Minor, in abstract: "These cameras inherently capture movement and depth information in events" - I would argue that event cameras are similar to classic ones in captureing depth. They provide continuous 'tracking', but the depth is not directly measured. At the very least, the advantage is not obvious.

2) In event-based processing it is specifically important to understand how exactly the events are fed into the DNN, and what are the implications of the approach / which other similar methods exist in literature. The Algorithm 1 answers this to a degree, but I am not sure I understand if the temporal window is always fixed, and if the number of the events within this window is subsampled to a constant value. What would happen if there are fewer events than Np (1024) due to the lack of motion?

---

> ### Author Response · Authors · 2023-11-13
> **We are grateful for your acknowledgment of our paper presentation and for your valuable comments regarding the “notable contribution” of PEPNet. We will open source the PEPNet code as soon as our paper is accepted. We hope that event-based SLAM based on Point Clouds will get more attention.**
>
> **Weakness 1.  & 2. More literature review and dataset evaluations.**
>
> Thank you sincerely for providing the two references and datasets. Upon thorough review, we have incorporated a description and citation of these in our revised paper in line 34 and 110. Nevertheless, we would like to highlight the differences: the references provided  are for motion segmentation, while our work is focused on 6 DOFs.  Motion segmentation is associated with a higher-order semantic task which is different from a straight regression task. For motion segmentation, an elaborate decoder module is required. The MVSEC and EV-IMO datasets have significantly enriched the field of event cameras. We have initiated an extensive evaluation of PEPNet across a broader spectrum of datasets. However, due to the long duration of the whole experimental process and training, we aspire to present PEPNet's performance at the earliest opportunity.
>
> Admittedly, in the field of CPR, there is very limited dataset. This is because 6DoFs with event camera is still an emerging field. We are aware of this limitation, and we hope to contribute in the future to this issue. Nevertheless, the field of 6DoF has already gained a lot of attention, such as ref  [1],[2],[3]. Our experiment is designed to compare in detail with all these references. For instance, we have analyzed the model’s performance on random split, novel split and error distribution.
>
> **Weakness 3. It would benefit the method if the loss function incorporated some geometry constraints.**
>
> Integrating geometric constraints into the loss function will enhance the network's ability to learn low-level geometric features. This requires extensive research and long preparation, and the combination of geometry and network could then be a complete  work [4] [5]. Consequently, we are committed to further improving our network capabilities based on the valuable feedback and insights you provide. In this paper we are more focused on the architecture of how to handle event clouds using point cloud processing networks.
>
> **Weakness 4. On motion estimation problem, with a random split, it is highly likely the network overfitted.**
>
> This is a very insightful observation. RANDOM split leads to overfitting problems, but NOVEL split was proposed by SP-LSTM [1] to address this problem. Subsequent work has been carried over to evaluate the generalization ability of the model on these two different methods. The MSE performance shows that our PEPNet achieves SOTA results not only on random split but also on the more challenging novel split demonstrates the generalizability of our proposed model.
>
> **Q1. Event cameras are similar to classic ones in capturing depth.**
>
> We apologize for the misunderstanding due to our unsuitable presentation. We agree that the events generated by the event camera provide continuous tracking, and that the depth of an object in a known scene with respect to the camera can be expressed in terms of the number of events generated by its motion. In more detail, for a pentagram in the shape sequence, moving farther away from the camera generates fewer events, and the opposite generates more events. We would amend this sentence to “These cameras implicitly capture movement and depth information in events.”
>
> **Q2. What would happen if there are fewer events than Np (1024) due to the lack of motion?**
>
> The temporal window in PEPNet is not fixed. It is processed whenever a sufficient accumulation of 1024 points occurs. In the datasets we utilize, simple scence typically yield around 5k events in 5 milliseconds, while complex scenarios generate between 10k to 20k events within the same time interval. This is often enough for datasets with a resolution of 5ms for the GT 6DOFs pose. In cases where the number of accumulated points falls short of 1024, we can employ a repeat sampling method (bootstrap resample) to augment the input to the network, ensuring that this adjustment has no impact on the final network output.
>
> [1] Nguyen A, Do T T, Caldwell D G, et al. Real-time 6dof pose relocalization for event cameras with stacked spatial lstm networks[C] CVPR2019
>
> [2] Jin Y, Yu L, Li G, et al. A 6-DOFs event-based camera relocalization system by CNN-LSTM and image denoising[J]. Expert Systems with Applications, 2021.
>
> [3] Lin H, Li M, Xia Q, et al. 6-DoF Pose Relocalization for Event Cameras With Entropy Frame and Attention Networks[C]// ACM SIGGRAPH 2022.
>
> [4]lRelative geometry-aware siamese neural network for 6dof camera relocalization[J]. Neurocomputing, 2021
>
> [5] Back to the feature: Learning robust camera localization from pixels to pose[C] CVPR2021

---

### Author Response · Authors · 2023-11-16

We extend our sincere appreciation to the three reviewers for their valuable and high-quality reviews. Your constructive insights have been instrumental in refining our experiments and enhancing the overall structure of PEPNet, resulting in a notable improvement in its quality.

We are delighted to highlight the unanimous recognition from all reviewers regarding PEPNet’s pioneering approach, which seamlessly integrates point cloud and event-based CPR tasks. The accolades of “**notable**,”“**insightful**,” and “**better performance**,” bestowed upon our work truly reflect its groundbreaking nature.